# Stereotactic Radiotherapy to the Prostate and Pelvic Lymph Nodes for High-Risk and Very High-Risk Prostate Cancer in a Setting with a Hydrogel Spacer: A Toxicity Report

**DOI:** 10.3390/cancers17121970

**Published:** 2025-06-13

**Authors:** Elisha Fredman, Roi Tschernichovsky, Danielle Shemesh, Miriam Weinstock-Sabbah, Ruth Dadush Azuz, Roman Radus, Assaf Moore, Dror Limon

**Affiliations:** 1Department of Radiation Oncology, Davidoff Cancer Center, Rabin Medical Center, Petah Tikvah 494149, Israel; roits@clalit.org.il (R.T.); daniellekl@clalit.org.il (D.S.); miriamsa3@clalit.org.il (M.W.-S.); ruthda1@clalit.org.il (R.D.A.); romanrad@clalit.org.il (R.R.); drorl@clalit.org.il (D.L.); 2Department of Molecular Cell Biology, Weizmann Institute of Science, Rehovot 763270, Israel; 3Department of Radiation Oncology, Memorial Sloan Kettering Cancer Center, New York, NY 10065, USA; assafmoore@gmail.com

**Keywords:** prostate, radiotherapy, pelvic nodes, stereotactic, peri-rectal spacer, toxicity

## Abstract

As the most common non-skin malignancy in men, prostate cancer can now be treated using modern techniques that substantially reduce the number of fractions to as few as five. Data is beginning to emerge to support application of this efficient method when additionally treating the pelvic lymph nodes in patients with higher-risk disease. A concern, however, is the risk of increased side effects when treating this larger volume, especially on bowel function. A proven method to decrease bowel toxicity is placement of a peri-rectal spacer to displace the rectum away from the prostate, thereby decreasing radiation exposure. Thus far, no studies have presented a toxicity profile for this treatment method in a setting with a spacer. In this 100-patient series, we demonstrate that five-fraction pelvic radiotherapy can be carried out safely and that bowel-related side effects are notably reduced when a spacer is used, providing evidence for optimal radiation for these patients.

## 1. Introduction

Prostate cancer is one of the most common cancer types among men worldwide [1]. Due to its uniquely low α/β ratio and the resulting radiosensitivity relative to the surrounding tissue [2], multiple studies of moderately hypofractionated [3,4,5] and ultra-hypofractionated [6,7] radiotherapy regimens have demonstrated the potential to substantially reduce the number of treatments that are necessary. Most recently, the Prostate Advances in Comparative Evidence-B (PACE-B) trial demonstrated the non-inferiority of five-fraction prostate stereotactic radiotherapy (SABR) compared with classically fractionated and moderately hypofractionated regimens [7].

While the above studies included patients with lower-risk disease who required radiation to the prostate alone, more recent data supports the added benefit of electively including the draining pelvic lymph nodes (ENI) for those with more aggressive clinically high-risk disease [8]. POP-RT was a randomized phase III trial comparing prostate-only with prostate-plus-pelvic nodal radiation using moderate hypofractionation. It found improved biochemical failure-free survival and disease-free survival in the lymph node arm. Older trials failed to show a similar benefit, but they employed lower doses of radiation and shorter durations of androgen deprivation therapy (ADT), both of which are now known to be insufficient [9,10]. Most recently, the SHARP consortium demonstrated very high levels of pelvic control and biochemical failure-free survival in a pooled database of patients receiving five-fraction SABR to the prostate including ENI [11].

Delivering SABR concurrently to the prostate and pelvic lymph nodes, however, raises concerns of increased genitourinary (GU) and gastrointestinal (GI) side effects due to the proportions of the bladder, large bowel, and small bowel that are exposed to high-dose radiation. Data is only beginning to emerge regarding the safety of SABR to the prostate with ENI, including a hospital-based cohort study of treatment trends from the National Cancer Database [12] showing a slight increase in its use; a 30-patient feasibility study that found increased rates of GU and GI toxicity [13]; and a pooled analysis that included 60 patients who received SABR and found increased grade 2 toxicity, though not grade 3 [14], as well as the above-mentioned SHARP data. The same SHARP consortium previously published a pooled series of phase 2 trials and registries within which 66 patients had SABR including ENI [15]. Larger, updated cohort studies and real-world data are, therefore, still needed.

With regard to local toxicity from prostate radiotherapy, an additional tool that has emerged and grown in popularity is peri-rectal spacer insertion. Whether in the form of a resorbable polyethylene glycol (PEG) or hyaluronic gel or a manually inflated balloon, insertion of a physical barrier between the prostate and rectum prior to the start of radiotherapy has been demonstrated in multiple studies to significantly reduce the rectal dose and minimize short- and long-term toxicity [16,17,18,19,20]. While initially applied in cases of conventional and moderately hypofractionated radiation, further experiences have demonstrated a benefit regarding rectal toxicity in the setting of SABR as well [21]. As yet, however, there exists little published data regarding the potential toxicity benefits of peri-rectal spacing when SABR is administered to a substantially larger radiation field, including a greater degree of exposure to the large and small bowels in the context of ENI for high-risk patients.

In the context of increasing interest in the application of SABR with ENI for high-risk prostate cancer, we present safety and early efficacy outcomes for patients with high-risk prostate cancer treated at our institution with five-fraction SABR including ENI in a setting with peri-rectal hydrogel spacing.

## 2. Materials and Methods

### 2.1. Participants

Internal review board approval was obtained to review a prospectively collected series of all patients ≥ 18 years old with high-risk and very high-risk National Comprehensive Cancer Network disease categories treated at our center with five-fraction SABR including ENI between 2021 and 2024 (beginning when this modality was first introduced). All patients had Eastern Cooperative Oncology Group performance statuses of 0–2, and all cases included at least one of the following: radiographic (3T-MRI) T3–4 disease, a Gleason score ≥ 8, PSA ≥ 20 ng/mL, and/or PET-PSMA-positive lymphadenopathy below the bifurcation of the common iliac vessels [22]. Those with an international prostate symptom score (IPSS) > 15 or an MRI-based prostate volume > 80 cc were excluded. All patients were prescribed 18–24 months of ADT, with or without the addition of Abiraterone, as per standard practice.

### 2.2. Treatment

ADT was initiated at least one month prior to the start of SABR, comprising 2–4 weeks of a non-steroidal ARi (50 mg of Bicalutamide) and injection of a GnRH agonist (Triptorelin) midway through, in either a 3- or 6-month formulation. All patients without physical or radiographic findings of posterior extracapsular extension (ECE) were recommended insertion of a polyethylene glycol hydrogel spacer (SpaceOAR^TM^, Boston Scientific, Marlborough, MA, USA) and gold fiducial markers, while fiducial markers alone were recommended for those with posterior ECE. Briefly, the procedure was performed under sterile conditions using local anesthesia in the department of radiation oncology. Both the spacer and the fiducials were inserted transperineally under trans-rectal ultrasound guidance. Proper deployment and placement of the three fiducials in the bilateral prostate and a minimum of 10 mm of posterior displacement of the rectum by the spacer at center-midgland were confirmed by direct visualization using ultrasound. The patients were prescribed prophylactic antibiotics to minimize infection risk, comprising three days of fluoroquinolone (allergy permitting) given twice daily beginning the morning of the procedure.

During the week after the procedure, CT and 3T-MRI simulation scans were performed with an empty rectum and a comfortably full bladder. Standard immobilization techniques, including knee and ankle supports, were used.

Treatment planning was based on CT/MRI fusion, with MRI data contributing especially to the contouring of the prostate gland and prostatic urethra. The prostate clinical target volume (CTV) and the seminal vesicles were expanded by 5 mm, limited to 3 mm in the posterior direction. The elective pelvic LN volume was delineated as an anatomically limited 7 mm contour around the pelvic vessels as per RTOG contouring guidelines [23] and expanded by an additional 5 mm. The relevant surrounding normal organs at risk (OARs), including the bladder, rectum, urethra, large and small bowel, penile bulb, and pelvic bones, were contoured based on the CT simulation data as per standard practice. Three dose levels were prescribed for the planning target volumes (PTVs): the prostate received 40 Gy (79%), 37.5 Gy (1%), or 36.25 Gy (20%); the seminal vesicles received 35 Gy; and the pelvic LN received 25 Gy. The patients could receive a simultaneous integrated boost (SIB) to a dominant intraprostatic lesion (DIL) up to 40–45 Gy dependent on pathology/imaging concurrence and the ability to maintain normal tissue constraints. The dose to the OARs was calculated using the ARIA^®^ treatment planning system and carefully analyzed based on accepted dose constraints through dose–volume histogram representation and review. Radiation was prescribed for homogeneous coverage of the three PTVs and the CTV of the prostate gland (Table 1). Treatment was delivered under cone-beam CT image guidance aided by the presence of the intraprostatic fiducials to assure precise target coverage and consistent OAR avoidance. Radiation was administered every other day over 1.5–2 weeks.

Baseline urinary and bowel functions were assessed at initial presentation, every three months following the completion of radiation for the first year, and every six months thereafter. GU and GI toxicity were quantified based on Common Terminology Criteria for Adverse Events (CTCAE) version 5. Patient-reported quality of life (QoL) specifically in the urinary and bowel domains was collected based on the Expanded Prostate Cancer Index Composite (EPIC) 26 at baseline and at three months and one year. For the purposes of this report, toxicity and QoL endpoints were assessed between three and twelve months post-radiation. While they are early given the relatively limited follow-up and the context of ADT, PSA results are reported, but they are not considered a primary focus of this analysis.

### 2.3. Statistical Analysis

Patient demographics and tumor characteristics were analyzed using medians and interquartile ranges (IQRs) for continuous variables and percent proportions for categorical and non-continuous variables. Descriptive statistics were used to describe the rates of GU and GI toxicity by grade and the proportions of patients with minimally clinically important changes (MCICs) on the EPIC questionnaire [24]. For grade 2 toxicity estimates, binomial exact confidence intervals (CIs) were calculated.

The PSA response was reported as the change in the absolute quantity from the baseline value tested within one month prior to study enrollment, with presentation of the initial acute response three months post-SABR and at twelve months.

Univariable and multivariable logistic regressions were used to assess associations between age, prostate volume, and baseline IPSS and GU and GI toxicity, as well as between the presence of a hydrogel spacer and GI toxicity, without a priori exclusion of any factors in the multivariable analysis. *p* < 0.05 was considered statistically significant.

## 3. Results

The median patient age was 73 years (IQR: 69–77 years; Table 2) and the median pretreatment PSA was 12.1 ng/mL (IQR: 7.4–21.8). In total, 69 patients had grade group 4 or 5 disease and 72 were in stage group IIIA-IVA. MRI-based T3a/3b disease was noted in 25 and 15 patients, respectively. The patients with clinical T1c disease had one or more additional high-risk features of grade group 4 or 5 pathology, radiographic N1 involvement, and/or PSA above 20. One patient had Gleason 6 pathology on biopsy, though his PSA was consistently elevated above 20 and his MRI suggested possible ECE. All patients had PET-PSMA systemic staging prior to treatment onset, and radiographic nodal involvement was found in seven cases. The median prostate volume at the time of MRI simulation was 39 cc (IQR: 30–50). In total, 98 of the 100 patients were treated with neoadjuvant/concurrent/adjuvant ADT, 74 prescribed for 18 months and 24 prescribed for 24 months. In addition, 96% of the patients underwent placement of three gold fiducial markers, and 70% also had a hydrogel spacer placed, undergoing the procedure 6–9 days before simulation. All fiducial and hydrogel spacer procedures were completed without complications or post-procedural toxicity, and there were no associated infections documented. In total, 32 patients received an SIB to a DIL (45 Gy or 40 Gy when receiving 40 Gy or 36.25–37 Gy in total for the gland, respectively), and all PSMA-positive nodes were given an SIB to 30–32 Gy.

With a median follow-up of 20 months (range: 12–36), grade 1 GU side effects were reported in 28% at three months, 18% at six months, 11% at nine months, and 5% at twelve months. Grade 2 GU side effects, all of which were due to the need to prescribe alpha-antagonist medication, occurred in 22% at three months, 11% at six months, 6% at nine months, and 3% at twelve months. One instance of grade 3 urinary obstruction occurred within one month of completing SABR, which resolved by three months, for a total 6–12-month grade 2+ rate of toxicity of 14% (95% CI: 7.9–22.4%). No grade 4 toxicities were reported. Grade 1 GI toxicity was reported in 14% of patients, decreased to 8% by six months, and resolved by nine months. One instance of grade 2 GI toxicity (frequency and tenesmus) was reported at the six-month follow-up and resolved by nine months. No grade 3+ GI toxicities occurred, for a total grade 2+ rate of 1% (95% CI: 0.03–5.4%). Among the 70 patients who underwent placement of a hydrogel spacer, three experienced a grade 1 GI toxicity (4.3%) vs. 11/30 reporting any GI toxicity among the non-spacer cohort (36.7%). No differences in reported toxicities were observed among those who did or did not receive an SIB.

Dosimetrically, the presence of a hydrogel spacer was associated with a reduction in the volume of the rectum receiving both high-dose and intermediate/integral-dose radiation. This was consistently seen across binned dose levels from the volume of the rectum receiving 95% of the prescribed dose (V95%) through 50% of the prescribed dose (V50%) (Figure 1).

With regard to the measures of acute changes in QoL, at the three-month follow-up, 5%, 18%, and 4% of the patients reported MCICs in the urinary incontinence, urinary obstructive, and bowel domains, respectively (Figure 2). At the twelve-month follow-up, the reported rates were 2%, 3%, and 0%. No particular patient prognostic factor was found to significantly influence GU toxicity. Patient-reported CTCAEv5 GI toxicity was statistically inversely associated with the presence of a hydrogel spacer (Table 3), showing a decreased risk of any GI toxicity when a spacer was inserted.

At the three-month follow-up, the mean and median PSA had decreased from 24.0 ng/mL to 0.15 ng/mL and 12.1 ng/mL to 0.01 ng/mL, respectively (range: 0–4.5), with the mean PSA further decreasing to 0.09 ng/mL by twelve months and the median PSA unchanged at 0.01 ng/mL. One of the two patients who refused ADT had a PSA decrease of 39% at three months, which then rose back to pretreatment levels by twelve months, and his PET PSMA was suggestive of a distant bone recurrence. As such, the one-year progression-free survival (PFS) and overall survival (OS) were 99% and 100%, respectively.

## 4. Discussion

Given its persistently high incidence, improving prostate cancer treatment efficiency while maintaining oncologic outcomes is a critical goal. However, as long-term survival is common, advancements in treatment techniques must emphasize reducing toxicity for optimal quality of life, a value that cannot be sacrificed in exchange for a shorter treatment course. In our experience with SABR with ENI for high-risk prostate cancer in the presence of a peri-rectal hydrogel spacer, we found overall low rates of both acute and intermediate-term GU and GI toxicity, and the presence of a spacer was associated with even lower risk. Dosimetrically the presence of a hydrogel spacer was associated with a substantial reduction in radiation exposure to the rectum, similar to findings observed in a pivotal hydrogel spacer trial in a setting with conventional fractionation [16].

These overall toxicity rates are similar or favorable in comparison to recent prospective trials of moderately hypofractionated prostate-only radiation [3,4,5], prostate-only SABR in lower-risk patients [6,7], and emerging reports of five-fraction pelvic SABR in high-risk settings [14,25]. Regarding moderate hypofractionation, the PROFIT trial reported acute and late grade 2/3 GU toxicity rates of 27%/3.9% and 20%/2%, respectively, and rates of acute and late grade 2/3 GI toxicity of 16%/0.7% and 7.4%/1.5%, respectively. In the CHHiP trial, high rates of acute grade 2+ GU toxicity in the 60 Gy arm (49%) decreased to 5% during long-term follow-up. In comparison, a recent prospective trial of prostate-only SABR, PACE-B, reported acute worst RTOG grade 2 or more severe GI side effects of 10% and GU side effects of 23%. In total, 16% of subjects had GI side effects of CTCAE grade 2 or worse in the acute setting, significantly more than in the hypofractionation arm, but this finding lost significance by twelve weeks post-radiation. Peri-rectal spacers were not utilized in the trial [7].

In a recently published pooled analysis that included patients who underwent SABR including pelvic radiation, Glicksman et al. reported a higher rate of grade ≥ 2 GU toxicity in the SABR arm at three months of 56% vs. 30% among moderately hypofractionated subjects [14]. Our results are very similar to those of both regimens when treating only the prostate. Importantly, GU toxicity continued to diminish over time in both arms, a phenomenon similarly seen among our cohort, with rates of grade 1 and 2 GU toxicity of 5% and 3% by one year after SABR, respectively.

PACE-B reported a peak grade 2 urinary frequency of 10% at fifteen months in the SABR arm [26], which was not seen in this cohort. This may have been due to the relative regularity with which the patients were treated with chronic alpha-adrenergic medication or local/regional practice trends. It has been reported in early trials of two-fraction prostate SABR that acute grade 2 GU toxicity tends to be largely due to prescribing alpha-adrenergic antagonist or beta-3 adrenergic agonist medications [27], which can be influenced by regional practices and norms.

Regarding GI toxicity, Glicksman et al. reported a grade ≥ 2 GI toxicity rate of 11.7% at three months, which was numerically but not statistically higher than that in the moderately hypofractionated cohort (5.7%). Among the patients included in the SHARP consortium analysis, acute and late grade ≥ 2 GI toxicity were reported in 5% and 9%, respectively, and late grade 3 GI toxicity occurred in 0.9% of the patients [15]. In our cohort of 100 patients, 14% reported acute grade 1 GI toxicity that resolved by the nine-month follow-up, and one instance of transient grade 2 GI toxicity emerged at six months, similarly resolving by nine months. Among those with a peri-rectal spacer, only 4.3% experienced an acute grade 1 GI side effect. The favorable GI toxicity profile in our cohort was likely due to the placement of a peri-rectal spacer in most patients, the presence of which was found to be associated with a lower risk of low-grade GI toxicity. This finding is consistent with previous reports of the benefit of peri-rectal spacing in the setting of prostate-only radiotherapy of various dose fractionations. Of note, there were no reported complications resulting from the hydrogel insertion procedure, a concern, although rare, that has been raised in recent big-data analyses [28].

There are currently a variety of methods used in clinical practice for peri-rectal spacing, which utilize different materials applied via slightly different injection methods. The three most common materials in use, for which there are supporting data in the literature, are PEG hydrogel (used in this cohort), hyaluronic acid-based gel, and an implanted balloon that is locally inflated. Numerous studies, both prospective and retrospective, as well as systemic reviews, have demonstrated the ability of hydrogel-based spacers to significantly improve rectal dosimetry and decrease GI toxicity [16,17]. Notably, this was demonstrated specifically in the context of high-dose ultra-hypofractionated SABR [21]. Data also support the use of hyaluronic acid-based spacers, which were initially shown to decrease radiation doses in the rectum in the context of brachytherapy [29] and were more recently studied in the context of conventionally fractionated [19] and moderately hypofractionated [30] radiotherapy, where rates of intermediate-grade toxicity were shown to improve in the context of the spacer. Although the data are more limited, placing a self-absorbing peri-rectal balloon spacer has also been shown to reduce radiation doses in the rectum to a degree similar to gel spacers [31]. We observed significant decreases in both the rectal dose and GI toxicity among those who had a spacer in our experience with five-fraction SABR with ENI.

Careful patient selection is an important factor in the safe delivery of SABR with ENI to minimize short- and long-term side effects. In the above-mentioned analysis from Glicksman et al., being older was a significant risk factor for developing late grade ≥ 2 GI toxicity. The strategic use of a peri-rectal spacer, fiducial marker-based image guidance, and careful attention to rectal dose constraints are likely important factors mediating the potential impact on GI function.

Owing to the known dose–response radiobiologic properties of prostate cancer, the FLAME [32] and hypo-FLAME trials [33] previously reported the relative safety and improved biochemical disease-free survival advantages of a focal boost to the DIL. Though not based on a priori randomization, we observed no differences in toxicity between patients who did or did not receive an integrated boost. Based on the iso-toxic and excellent long-term oncologic outcomes observed in hypo-FLAME [34], further investigation of the role of focal dose escalation is supported.

As nearly all patients in this cohort received long-term ADT, conclusions regarding treatment efficacy were felt to be premature. Nonetheless, we reported mean and median PSA values of 0.09 and 0.01 ng/mL at one year post-treatment, for a biochemical progression-free survival rate of 99%. Notably, there were no reported recurrences in the treated pelvic nodes. Our findings are consistent with recent similar studies of SABR with ENI in high-risk patients, in particular a study reported by the SHARP consortium [11], where with a median of 51 months of follow-up they recently reported a PFS of 89% and a 9.9% rate of distant metastases as the site of failure. As in our experience, there were almost no recurrences at one year. In addition, they demonstrated the effectiveness of this 25 Gy pelvic dose regimen, finding overall pelvic control of 98.2%. By the one-year follow-up, we observed no pelvic nodal failures. Continued follow-up will be necessary to analyze long-term oncologic outcomes.

Currently, there are multiple accruing prospective trials investigating the relative feasibility and utility of SABR with ENI in the setting of high-risk prostate cancer compared to other fractionation schemes, including the PRIME trial [35] and NRG GU013 (NCT05946213), both comparing SABR to a more protracted regimen using a non-inferiority design. PACE-NODES (NCT05613023) will compare prostate-only SABR and prostate with ENI in high-risk patients to assess safety and efficacy. In a somewhat different respect, ASCENDE-SBRT will compare the superior progression-free survival achieved with brachytherapy-based extreme dose escalation demonstrated in ASCENDE-RT [36] with pelvic SABR, noting that the SABR with ENI arm will be experimental. This study is of particular interest, as the advantages of brachytherapy in ASCENDE-RT came at the cost of increased long-term toxicity [37] and perhaps SABR with ENI can achieve a more favorable balance.

The strengths of this study include analyzing a prospective cohort representing real-world application of SABR with ENI in this population, the relative homogeneity of the baseline patient and tumor characteristics, and the high degree of consistency across the treatments within the cohort (dose, fractionation, the use of a hydrogel spacer and fiducials, and ADT). To the best of our knowledge, this represents the first report of a cohort of high-risk patients on this scale undergoing SABR with ENI in the presence of a peri-rectal spacer. Important limitations, however, include not being a prospective trial with prespecified endpoints, a lack of a direct comparison with a standard treatment arm, and limited follow-up precluding our ability to report meaningful oncologic outcomes. As prospective trials of prostate SABR with ENI accrue, some of which allow for peri-rectal spacing techniques, these results are both hypothesis-generating and supportive of the relevance of this treatment technique and the importance of continued investigation.

## 5. Conclusions

In conclusion, in this cohort of 100 patients with high-risk and very high-risk prostate cancer treated with five-fraction SABR with ENI in the context of a peri-rectal spacer, treatment was well tolerated with limited reported acute and intermediate- term GU and GI toxicity, in particular among those with a spacer. Prospective and comparative data are awaited from the currently accruing clinical trials to further support this radiotherapy technique in the high-risk population.

## Figures and Tables

**Figure 1 cancers-17-01970-f001:**
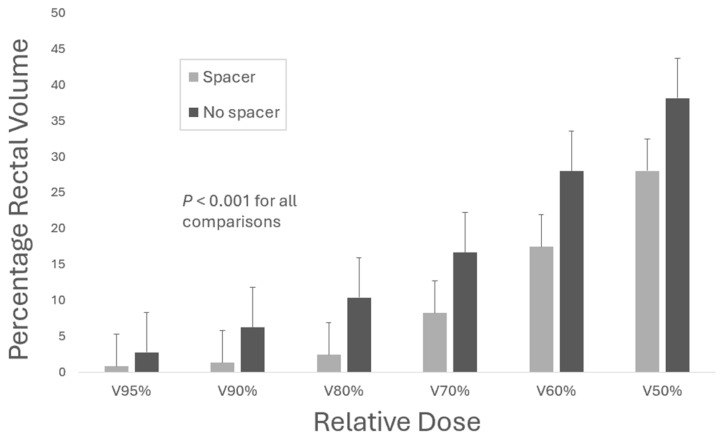
A dosimetric comparison of the radiation exposure to the rectum, presented as the percentage of the prescribed dose overlapping with a percentage of the rectal contour (V-percent dose), between spacer and non-spacer patients.

**Figure 2 cancers-17-01970-f002:**
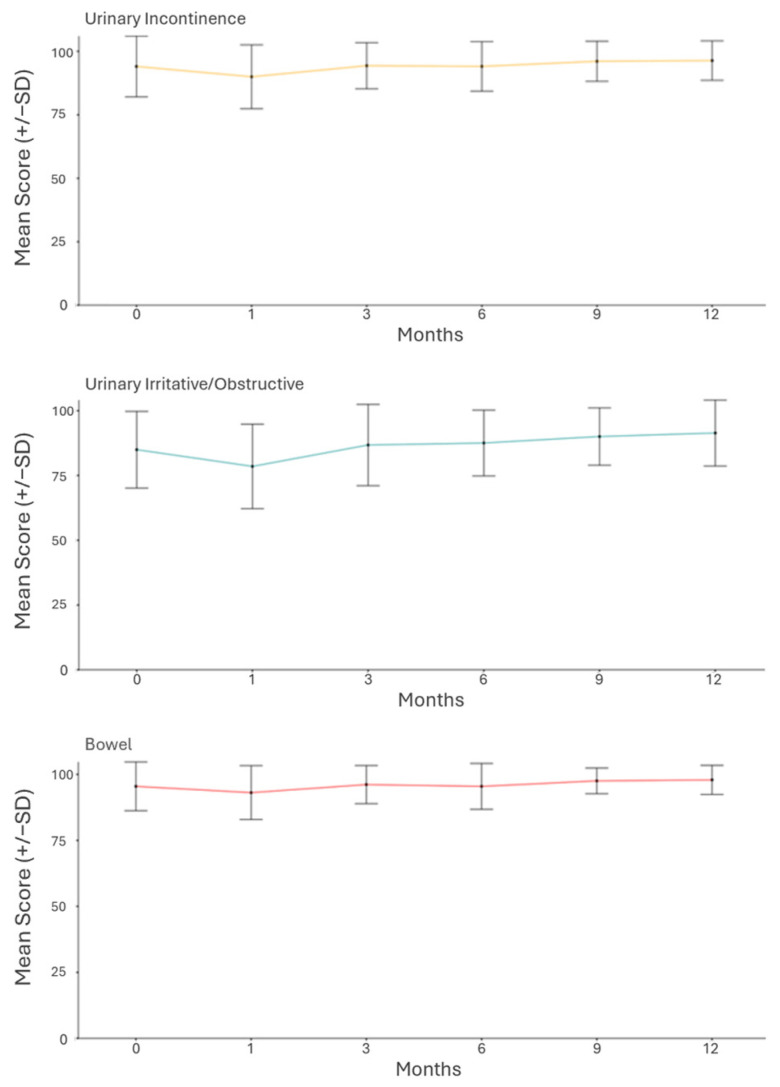
Changes in patient-reported quality of life based on EPIC-26 domains: urinary incontinence, urinary irritative/obstructive, and bowel.

**Table 1 cancers-17-01970-t001:** Prescribed radiotherapy target coverage.

Target	Volume (%)	Dose (%)	Prescription
PTV of prostate	95	99	36.25–40 Gy
CTV of prostate (gland)	99	99	36.25–40 Gy
PTV of seminal vesicle	95	99	35 Gy
PTV of nodes	95	95	25 Gy
PTV of boost (optional)	95	95	40–45 Gy

PTV—planning target volume; CTV—clinical target volume; Gy—gray.

**Table 2 cancers-17-01970-t002:** Patient characteristics.

		N = 100
Age		
	Median (IQR)	73 (69–77)
Clinical stage		
	T1c	40
	T2c	20
	T3a	25
	T3b	15
	N1 (among above)	7
PSA		
	Median (IQR)	12.1 (7.4–21.8)
ISUP grade group		
	GG1	3
	GG2	15
	GG3	13
	GG4/5	69
Systemic staging		
	PET PSMA	100
Stage group		
	IIB	8
	IIC	20
	IIIA	14
	IIIB	40
	IIIC	29
	IVA	7
Risk group		
	High	54
	Very High	46
Prostate volume (cc)		
	Median (IQR)	39 (30–50)
IPSS baseline score		
	Median (IQR)	7 (5–11)
SHIM baseline score		
	Median (IQR)	12 (6–16)

IQR—interquartile range; IPSS—international prostate symptom score; SHIM—sexual health inventory in men.

**Table 3 cancers-17-01970-t003:** Univariable and multivariable analyses of associations with genitourinary and gastrointestinal toxicity.

		**GU Toxicity**							
		**Grade 1**				**Grade 2**			
		**Univariable**		**Multivariable**		**Univariable**		**Multivariable**	
3 months	Variable	HR (95% CI)	*p*	HR (95% CI)	*p*	HR (95% CI)	*p*	HR (95% CI)	*p*
	Age	0.98 (0.92–1.05)	0.62	0.98 (0.92–1.05)	0.59	1.00 (0.94–1.08)	0.92	0.99 (0.93–1.07)	0.88
	Prostate volume	1.00 (0.98–1.03)	0.73	1.01 (0.98–1.03)	0.68	1.03 (0.99–1.06)	0.06	1.03 (0.99–1.06)	0.07
	IPSS	1.00 (0.88–1.14)	0.96	0.99 (0.88–1.14)	0.97	1.05 (0.91–1.20)	0.49	1.03 (0.89–1.18)	0.72
12 months	Variable	HR (95% CI)	*p*	HR (95% CI)	*p*	HR (95% CI)	*p*	HR (95% CI)	*p*
	Age	0.89 (0.77–0.99)	0.04	0.96 (0.82–1.13)	0.61	0.89 (0.78–1.01)	0.07	0.95 (0.81–1.13)	0.59
	Prostate volume	0.96 (0.92–1.04)	0.42	1.02 (0.96–1.09)	0.58	0.96 (0.93–1.05)	0.64	1.01 (0.95–1.08)	0.69
	IPSS	1.03 (0.79–1.33)	0.85	1.21 (0.85–1.73)	0.29	1.00 (0.75–1.34)	0.99	1.20 (0.83–1.73)	0.34
		**GI Toxicity**							
		**Grade 1**							
		**Univariable**		**Multivariable**					
	Variable	HR (95% CI)	*p*	HR (95% CI)	*p*				
3 months	SpaceOAR Yes vs. No	0.09 (0.27–0.35)	0.0004	0.08 (0.02–0.31)	0.0003				
	Age	1.01 (0.93–1.10)	0.82	0.98 (0.89–1.08)	0.68				
	Prostate volume	1.01 (0.98–1.04)	0.47	1.01 (0.97–1.04)	0.62				
	IPSS	1.06 (0.90–1.25)	0.46	1.02 (0.85–1.23)	0.80				
12 months	SpaceOAR Yes vs. No	NA	NA	NA	NA				

## Data Availability

Research data are stored in an institutional repository and will be shared upon request to the corresponding author.

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
