# Peer review of "Stereotactic Radiotherapy to the Prostate and Pelvic Lymph Nodes for High-Risk and Very High-Risk Prostate Cancer in a Setting with a Hydrogel Spacer: A Toxicity Report"

_cancers, 2025, doi:10.3390/cancers17121970_

Round 1
Reviewer 1 Report
Comments and Suggestions for Authors
I have read with great interest article entitled: Stereotactic radiotherapy to the prostate and pelvic lymph nodes for high- and very high-risk prostate cancer in the setting of a hydrogel spacer – a toxicity report. Authors analysed important topic within prostate cancer treatment: toxicity after placement of a peri-rectal spacer in context of giving high-dose prostate radiotherapy (including elective nodal irradiation). According to the obtained results presence of a hydrogel spacer resulted in reduced high- and intermediate-dose to the rectum and demonstrated a significant inverse association with short-term GI toxicity. This is a great article, however some questions need to be answered. First of all, authors should list and discuss other publications which investigated the use of different spacers in prostate cancer radiotherapy, demonstrating their effectiveness in reducing rectal radiation exposure and mitigating gastrointestinal (GI) toxicity (prospective studies, proton beam RT studies, data from large retrospective studies). Also, as authors stated maintaining oncologic outcomes is a critical goal. However, no outcomes with respect to PSA decline, PFS or OS were reported. Despite limited follow-up period, these end-points should be reported and compared with results from other prospective trials which included high-risk prostate cancer population (treated with either SABR or other hypofractionated regimens). This could serve as ‘’control’’ since there is no direct comparison arm within this single arm study. Please comment and compare results in discussion section.
Author Response
Comment 1: Authors should list and discuss other publications which investigated the use of different spacers in prostate cancer radiotherapy, demonstrating their effectiveness in reducing rectal radiation exposure and mitigating gastrointestinal (GI) toxicity (prospective studies, proton beam RT studies, data from large retrospective studies).
Response 1: Thank you for the comment. We agree with your suggestion and have added to the discussion section an additional paragraph (page 9, beginning line 276) dedicated to discussing other rectal spacers, including the two other primary materials widely commercially available, which have also been shown to be effective in general in reducing rectal toxicity in prostate cancer radiotherapy.
Comment 2: no outcomes with respect to PSA decline, PFS or OS were reported. Despite limited follow-up period, these end-points should be reported and compared with results from other prospective trials which included high-risk prostate cancer population (treated with either SABR or other hypofractionated regimens). This could serve as ‘’control’’ since there is no direct comparison arm within this single arm study. Please comment and compare results in discussion section.
Response 2: Thank you for the comment. The PFS and OS outcomes have been elaborated upon, both in the results section (page 8, beginning line 218), and in the discussion section (page 10, beginning line 308).
Reviewer 2 Report
Comments and Suggestions for Authors
cancers-3673359
I thank the editors for the opportunity to review the article by Fredman et al.
The article is well thought out and well designed. The introduction is short, which introduces the reader to the problems and issues related to prostate cancer and prostate stereotactic radiotherapy. The project has the approval of bioethics committees.
The selection of research methodology and research group is appropriate.
The presented results and discussion also do not raise any objections.
As a reviewer, I would like to draw your attention to a few issues:
I suggest expanding the abbreviation PACE as Prostate Advances in Comparative Evidence
Additionally, I propose to improve the quality and readability of the charts. The font is too small
In my opinion, the manuscript (after revision) will be ready for publication.
Author Response
Comment 1: I suggest expanding the abbreviation PACE as Prostate Advances in Comparative Evidence
Response 1: Thank you. The abbreviation has been expanded the first time it appears and only used in short form thereafter.
Comment 2: Additionally, I propose to improve the quality and readability of the charts. The font is too small
Response 2: Thank you for the suggestion. We have reformatted the charts and enlarged the fonts
Reviewer 3 Report
Comments and Suggestions for Authors
The manuscript mainly presents the benefit of a radiopaque hydrogel spacer in 100 high-risk prostate cancer patients underwent stereotactic radiotherapy with elective nodes. The focus is quite interesting, but the following issues should be addressed:
- There is no detailed description of how to detect the radiation dose exposed to the rectum and the corresponding exposed volume under the insertion of the spacer.
- In exploring the association between the spacer insertion and gastrointestinal toxicity, other potential variables such as age, prostate grade, PSA level, and radiation dose should be considered.
- In Table 3, factors without statistical significance in the univariate analysis should not be included in the multivariate analysis.
- Line 277-278, the author mentions that older age is a significant risk factor for late grade ≥ 2 gastrointestinal toxicity, but there is no relevant data in the manuscript to support it.
- With a sample size of merely 100 patients, the terminology of "large cohort" appears to lack justification in this study.
- In Table 2, the presentation of the staging details need optimization. Particularly, the justification for classifying all 40 T1c-stage patients into high-risk or very high-risk requires being explained.
- Both Figures 1 and 2 lack sufficient clarity in the current form.
Author Response
We appreciated all of your thoughtful comments and have addressed each one carefully. Please see below the list of comments and responses.
Comment 1: There is no detailed description of how to detect the radiation dose exposed to the rectum and the corresponding exposed volume under the insertion of the spacer.
Response 1: Thank you for your comment. Standard practice for detecting dose exposure is through applying specific and detailed constraints on normal structures in the radiation prescription, performing complex inverse planning in the treatment planning system, and carefully analyzing with dose volume histograms. In the setting of prostate radiation, on occasion though in clinical practice rarely even in the most advanced practices, intra-treatment dose can be detected through inserted diodes or scintillating devices, but these are not standardly used. Rather, precise organ alignment is performed at each treatment based on patient set up and on-board imaging, and if the structures are matched to the original CT simulation, the previously calculated doses are known to be met. We have added some additional brief elaboration on the matter to add clarity (page 3, beginning line 123).
Comment 2: In exploring the association between the spacer insertion and gastrointestinal toxicity, other potential variables such as age, prostate grade, PSA level, and radiation dose should be considered.
Response 2: Thank you for the suggestion. We agree that these additional data are important to include and therefore the table has been expanded. We have added UVA and MVA for GI toxicity, though as there were no reported toxicities at 12 months and only 1 case of grade 2 toxicity, those columns which are included in the GU analysis are not included in the GI analysis since they don’t have data. Lastly, we did not do an analysis based on prescribed dose because the dose limits to the rectum were the same across all dose levels, so there aren’t comparison variables.
Comment 3: In Table 3, factors without statistical significance in the univariate analysis should not be included in the multivariate analysis.
Response 3: Thank you for the comment. We went back and forth a number of times regarding this specific statistical question. After reviewing other recent studies that similarly looked at factors associated with GU and GI toxicity after radiotherapy for prostate cancer, including the recent SHARP consortium analysis, and in the context of there being fairly few events overall, we felt that it was more statistically consistent to not exclude any factors from the MVA. Since there were few toxicity events overall and the sample size was moderate though not very large, we didn’t want to potentially miss any associations between factors that could have potentially emerged. Therefore, while the chances were small, we included all factors in the MVA to account for any and all possible associations. The associated line has been updated in the methods section.
Comment 4: Line 277-278, the author mentions that older age is a significant risk factor for late grade ≥ 2 gastrointestinal toxicity, but there is no relevant data in the manuscript to support it.
Response 4: Thank you for noticing this unclear line in the paper. We were referring to the Glicksman analysis that was mentioned in the previous paragraph, and contrasting it to our findings. We have modified that sentence to enhance the clarity so that it is now less confusing.
Comment 5: With a sample size of merely 100 patients, the terminology of "large cohort" appears to lack justification in this study.
Response 5: We appreciate the point and have therefore removed all references to it being a “large” cohort.
Comment 6: In Table 2, the presentation of the staging details need optimization. Particularly, the justification for classifying all 40 T1c-stage patients into high-risk or very high-risk requires being explained.
Response 6: Thank you for the comment regarding clarification of staging. As you know, the T stage is just one factor that makes up the overall risk level staging, and is defined as diagnosis originally found after an elevated PSA. While 40 of the patients were a T stage 1c, they additionally had high risk features of either gleason grade group 4 or 5, were potentially N1, and/or had a PSA above 20, all of which would increase the risk level to high or very high risk. We added additional clarification and explanation to address your suggestion (Page 4, beginning line 173).
Comment 7: Both Figures 1 and 2 lack sufficient clarity in the current form.
Response 7: Thank you for the suggestion. We have reformatted the charts and made them easier to read.